# Recent Applications of Antireflection Coatings in Solar Cells

**Chunxue Ji** [1,2], **Wen Liu** [1,2], **Yidi Bao** [1,2], **Xiaoling Chen** [1,2], **Guiqiang Yang** [1,2], **Bo Wei** [1,3], **Fuhua Yang** [1,2] **and Xiaodong Wang** [1,2,3,4,*] 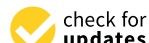

1   Engineering Research Center for Semiconductor Integrated Technology, Institute of Semiconductors, Chinese Academy of Sciences, Beijing 100083, China
2   Center of Materials Science and Optoelectronics Engineering, University of Chinese Academy of Sciences, Beijing 100049, China
3   School of Integrated Circuits, University of Chinese Academy of Sciences, Beijing 100049, China
4   Beijing Engineering Research Center of Semiconductor Micro-Nano Integrated Technology, Beijing 100083, China
*   Correspondence: xdwang@semi.ac.cn

**Abstract:** The antireflection coating (ARC) suppresses surface light loss and thus improves the power conversion efficiency (PCE) of solar cells, which is its essential function. This paper reviews the latest applications of antireflection optical thin films in different types of solar cells and summarizes the experimental data. Basic optical theories of designing antireflection coatings, commonly used antireflection materials, and their classic combinations are introduced. Since single and double antireflection coatings no longer meet the research needs in terms of antireflection effect and bandwidth, the current research mainly concentrates on multiple layer antireflection coatings, for example, adjusting the porosity or material components to achieve a better refractive index matching and the reflection effect. However, blindly stacking the antireflection films is unfeasible, and the stress superposition would allow the film layer to fail quickly. The gradient refractive index (GRIN) structure almost eliminates the interface, which significantly improves the adhesion and permeability efficiency. The high-low-high-low refractive index (HLHL) structure achieves considerable antireflection efficiency with fewer materials while selecting materials with opposite stress properties improves the ease of stress management. However, more sophisticated techniques are needed to prepare these two structures. Furthermore, using fewer materials to achieve a better antireflection effect and reduce the impact of stress on the coatings is a research hotspot worthy of attention.

**Keywords:** solar cell; antireflection coating; experiment data; characteristic matrix

## 1. Introduction

The solar cell is a device that directly converts light energy into electric energy by the photoelectric effect. Since the first solar cell was created in 1883, they have significantly progressed. Solar cells consisting of silicon are the first generation of solar cells, which are manufactured and used on a large scale and are divided into monocrystalline and polycrystalline silicon solar cells. Because there are many grain boundaries in polysilicon, which inhibit the continuous flow of electrons in semiconductors, the cell efficiency is lower than that of monocrystalline silicon solar cells. The second generation of solar cells is based on semiconductor thin film materials, such as GaAs (gallium arsenide), a-Si (amorphous silicon), CdTe (cadmium telluride), CIGS (copper indium gallium selenium), and CIS (copper indium selenium). Currently, the highest efficiency of Cd-free CIGS solar cells can reach approximately 23.35% [1]. In recent years, the third generation of solar cells has emerged, including heterojunction solar cells, tandem solar cells, quantum dot solar cells, organic polymer solar cells, dye-sensitized solar cells, hot carrier solar cells, and so on [2]. Compared with the first and second generations, its performance and stability are still limited, but it has enormous potential. Up to now, the maximum efficiency of

multijunction cells can reach 38% [1,2]. With the continuous development of solar cells, researchers are most concerned about their photoelectric conversion efficiency. However, one of the major bottlenecks limiting the efficiency of solar cells is the light loss due to surface reflection, inadequate transparency, and spectral mismatch, which accounts for nearly 7% of the decrease in solar cell efficiency [3], and the heat loss caused by high-energy charge carriers in the short wavelength range [4].

There are mainly two strategies to reduce reflection loss: (1) depositing single or multiple layer antireflection coatings or gradient refractive index thin (GRIN) coatings with matching optical properties on the substrate; (2) increasing the porosity of the material or etching the nanostructure array on the surface [5]. ARCs mainly use the principle of destructive interference to restrain the Fresnel reflection loss caused by the propagation of light in different media. Analyzing the optical and mechanical properties of different materials is of great significance for obtaining effective ARCs with outstanding performances. The idea of designing antireflection films is to approach the refractive index of the substrate coated with an antireflection film to the refractive index of the incident medium as much as possible without reducing the transmittance [6]. For GRIN thin films, gradually decreasing the refractive index from the substrate to the air could also ensure maximum light transmittance. Similarly, antireflection performance could also be obtained with porous materials and surface modifications. These methods could significantly improve photovoltaic conversion efficiency, thus enhancing the performance of solar cells.

This paper illustrates the theories and methods needed to design ARCs, including the equivalent interface, the characteristic matrix, the gradient refractive index, and the equivalent layer. Meanwhile, the characteristics of single, double, and triple layer antireflection coatings are briefly described and compared. After that, from the experimental aspect, the latest typical research on ARCs for solar cells in recent years is presented in three parts. In the first part, some widely used ARC materials and the applications of the corresponding materials on the surface of crystalline silicon solar cells are introduced. The second part is the research situation of ARCs applied to solar cells based on GaAs, a-Si, and other thin film materials. There are examples of preparing multiple layer coatings using the same material by adjusting the porosity or stoichiometric ratio of the materials to change the refractive index. The third part is related research on the third generation of high-efficiency solar cells, combining the advantages of the first and second generations of solar cells. For example, polymer-based antireflection film materials are used to achieve dual functions of antireflection and durability. These studies have further improved the efficiency of third generation solar cells. The optical and electrical parameters of solar cells and the materials of ARCs used in the cited literature are summarized at the end of each part for reference.

## 2. The Design Principle of Antireflection Coatings

### 2.1. Single Layer Antireflection Coating (SLARC)

The original way to reduce the surface reflection is by plating a low refractive index film on the substrate. With the help of the equivalent interface and characteristic matrix method, the reflectivity of the system after coating could be studied [7]. The two interfaces of a monolayer film could be mathematically represented by an equivalent interface (Figure 1a). The combination of the film and substrate could be regarded as a new substrate in which optical admittance could be used as the refractive index of the new substrate. The formula is as follows:

$$\begin{bmatrix} B \\ C \end{bmatrix} = \begin{bmatrix} cos\delta_1 & \frac{i}{n_1}sin\delta_1 \\ in_1sin\delta_1 & cos\delta_1 \end{bmatrix} \begin{bmatrix} 1 \\ n_s \end{bmatrix}$$

For the center wavelength, there are:

$$\delta_1 = \frac{2\pi}{\lambda} n_1 d_1 cos\theta_1 = \frac{\pi}{2}$$

$$Y = \frac{C}{B} = \frac{n_1^2}{n_s}$$

$$R = \left(\frac{n_0 - Y}{n_0 + Y}\right)^2 = \left(\frac{n_0 - \frac{n_1^2}{n_s}}{n_0 + \frac{n_1^2}{n_s}}\right)^2$$

**Figure 1.** The structures and equivalent interfaces of the (**a**) monolayer film and (**b**) multilayer film.

When the refractive indices satisfy:

$$n_1 = \sqrt{n_0 n_s}$$

Additionally, the optical thickness of the film is a quarter wavelength, and the reflectivity is zero. In the equations, $\delta_1$, $\lambda$, $\theta_1$ and $d_1$ refer to the phase thickness, center wavelength, incidence angle, and thickness of the film. $n_0$, $n_1$, $n_s$ are the refractive indices of the incident medium, film, and substrate, respectively. $Y$ is the optical admittance, and $R$ is the reflectivity. When the substrate is glass ($n_{\text{glass}} \approx 1.52$), the ideal refractive index of the antireflection film should be 1.23, but the lowest refractive index of the available material was 1.38 ($MgF_2$). Although it is not the ideal situation, after plating a single layer of $MgF_2$, the reflectivity at the center wavelength decreased from 4.2% to approximately 1.3%, and the transmission characteristics improved to some extent. As shown in Figure 2(a), it is impractical to achieve zero reflection with only a single antireflection film. Usually, a SLARC could reduce the reflectivity to approximately 2.5% in a wide spectral range of 450–1100 nm, and the residual reflection is too high for most applications.

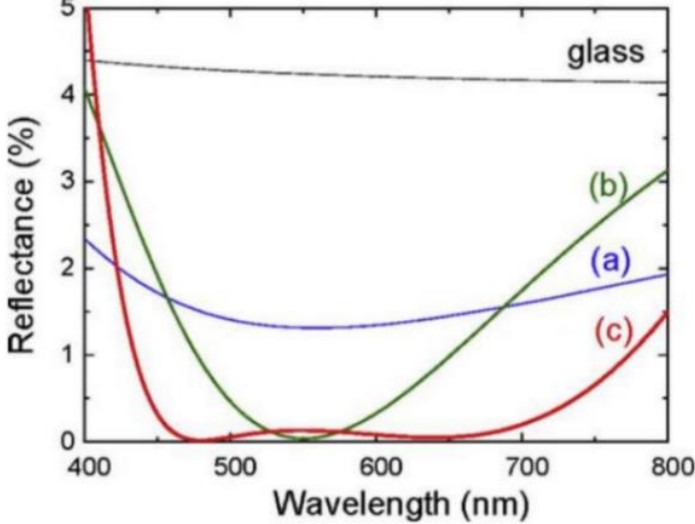

**Figure 2.** Reflectance curves of (a) SLARC, (b) DLARC, and (c) TLARC. Reprinted with permission [8]; 2010, Elsevier.

### 2.2. Double Layer Antireflection Coating (DLARC)

Generally, when the typical DLARC is used, the effective reflectivity decreases significantly compared with the SLARC, which approximates zero at the center wavelength and increases significantly when it deviates from the center wavelength (Figure 2(b)). The V-shaped reflectivity curve appears in the analyzed spectral range, for which the DLARC is also referred to as the V-coating. The structure of the double or multiple layer coating is depicted in Figure 1b. The second layer and substrate are equivalent to a new substrate:

$$n'_s = Y = \frac{n_2^2}{n_s}$$

Therefore, the bilayer system could be equivalent to the monolayer system. In other words, three interfaces were equivalent to one interface and then calculated in the same way as the monolayer system. When the thickness of each layer is a quarter wavelength, the condition for realizing zero reflectivity is as follows:

$$n_1 = \sqrt{n_0 n'_s} = \sqrt{n_0 \frac{n_2^2}{n_s}} = n_2 \sqrt{\frac{n_0}{n_s}}$$

$n_0$, $n_1$, $n_2$, $n_s$, $n'_s$ symbolize the refractive indices of the incident medium, the first layer film, the second layer film, the substrate, and the new substrate, respectively. When the material whose refractive index completely satisfies the above relationship cannot be found, it could be optimized by adjusting the thickness [9].

### 2.3. Triple Layer Antireflection Coating (TLARC)

The performance of the DLARC was better than that of the SLARC, which could only effectively reduce the reflection in a limited spectral range, and accordingly, using triple or multiple antireflection films to obtain lower reflectivity and a wider spectral width is requisite. A typical TLARC structure could be obtained by inserting a half-wavelength-thickness film into the middle of the V-shaped film with an equal thickness (quarter-half-quarter structure), which is also proved to be the most appropriate structure for the visible and far infrared regions [10]. As depicted in Figure 2(c), the insertion layer smooths the reflectivity curve and broadens the width of the low reflection band. Starting from the bottom layer, the equivalent substitution is carried out layer by layer, and finally, multiple interfaces are equivalent to one interface (Figure 2(b)). The equivalent admittance and reflectivity of the triple and multiple layer antireflection coatings could also be calculated with the characteristic matrix [7]:

$$\begin{bmatrix} B \\ C \end{bmatrix} = \left\{ \prod_{j=1}^{K} \begin{bmatrix} cos\delta_j & \frac{i}{n_j}sin\delta_j \\ in_j sin\delta_j & cos\delta_j \end{bmatrix} \right\} \begin{bmatrix} 1 \\ n_s \end{bmatrix}$$

$$\delta_j = \frac{2\pi}{\lambda} n_j d_j \cos\theta_j$$

$$Y = \frac{C}{B}$$

$$R = \left( \frac{n_0 - Y}{n_0 + Y} \right)^2$$

"j" is the sequence number of each layer, 1 denotes the top layer and K denotes the bottom layer. $n_j$, $\delta_j$, $\theta_j$ and $n_s$ are the refractive index, phase thickness, incidence angle of the "j" layer and substrate refractive index, respectively.

### 2.4. Gradient Refractive Index (GRIN) Coating

The GRIN coating is a special non-uniform ARC whose refractive index gradually changes along the vertical direction from the incident medium to the substrate. It could be regarded as a structure composed of a series of extremely thin films, which has the advantage of broadening the antireflection bandwidth and angle. If the thickness of each film is much smaller than the reference wavelength, the GRIN film could be regarded as an approximately continuous system without interfaces. For this reason, the reflection of light entering the substrate from the incident medium is very small [11]. However, when analyzing the limitations of the GRIN structure, it is a tedious task to match the refractive index of the thin film with that of the air and substrate. Perfect matching would be impossible, and there would always be some optical loss [5]. At the same time, the mechanical stress and durability of the multilayer film stacks cannot be ignored.

### 2.5. Equivalent Layer of the Symmetrical Film System

Apart from the ARCs designed with the characteristic matrix and GRIN coatings, the idea of an equivalent layer was also introduced and used for symmetrical film systems. In 1952, Epstein first analyzed the periodic symmetric film system mathematically and put forward a relatively sound concept of the equivalent refractive index [12]. For non-absorbing media, the characteristic matrix of a monolayer film is:

$$M = \begin{bmatrix} cos\delta & \frac{i}{\eta}sin\delta \\ i\eta sin\delta & cos\delta \end{bmatrix} = \begin{bmatrix} m_{11} & m_{12} \\ m_{21} & m_{22} \end{bmatrix}$$

The characteristic matrix of a non-absorbing multilayer film is the product of the characteristic matrices of each monolayer film:

$$M = M_1 \cdot M_2 \dots M_K = \begin{bmatrix} M_{11} & M_{12} \\ M_{21} & M_{22} \end{bmatrix}$$

It can be seen from the formula of a monolayer film that $m_{11} = m_{22}$. Generally speaking, $M_{11} \neq M_{22}$, but for the multilayer system with the middle layer as the center and placed on both sides symmetrically, $M_{11} = M_{22}$ in the characteristic matrix. Thus, the monolayer film and the multilayer film could be connected mathematically. In other words, there is an equivalent layer mathematically [7]. In a film system with high, medium, and low refractive indices, the medium refractive index films could be substituted with symmetric films composed of the other two materials, which means lower reflectivity could be achieved with two materials compared with the traditional DLARCs [13].

Selecting a suitable material is the first step in designing and preparing an ARC. Researchers must take the chemical and physical properties of the material and the matching of the refractive index and stress into consideration. Silicon-based materials, including silicon dioxide ($SiO_2$), silicon nitride ($SiN_x$), and porous silicon (PS), have strong compatibilities with silicon solar cells and have been widely used in related fields. The research shows that when the thickness of $SiO_2$ is greater than 20 nm, the antireflection effect is weakened when the thin film with any thickness is plated [14]. Therefore, other materials could be combined to achieve the dual functions of antireflection and surface passivation [15,16]. $SiN_x$ has good oxidation resistance, corrosion resistance, and insulation properties, and its optical properties could be optimized according to needs by adjusting the stoichiometric ratio (x= [N]/[Si]) [17]. Hydrogen atoms can passivate bulk defects and grain boundaries in silicon materials, reduce surface recombination speeds, and improve $V_{OC}$ and $J_{SC}$ [18]. Thus, $SiN_x$:H (n=2.1 at 633 nm) has dual effects of passivation and antireflection. Researchers combined $SiO_2$ and $SiN_x$:H to obtain lower reflectance and observed color changes by adjusting the thickness [19,20]. PS could decrease reflectivity, reduce the surface recombination speed, expand the sensitivity region in the spectrum, and increase the photogenerated carriers [21,22], so it could be used as a candidate material for preparing

antireflection films. The refractive index of PS varies with porosity, which could be achieved by adjusting the current density in the electrochemical methods [23]. Metal-based materials play an indispensable role in the design and application of ARCs, mainly including titanium dioxide ($TiO_2$), zinc oxide (ZnO), zinc sulfide (ZnS), magnesium fluoride ($MgF_2$), indium tin oxide (ITO), and so on. These materials are often prepared into nanostructures such as nanowires [24] and nanorods [25]. It is also a common method to prepare DLARCs from the same material with different porosities [26]. $MgF_2$ is a low-refractive-index material, which is often used in combination with other materials to reduce reflection [27,28]. ITO has excellent conductivity and transmittance and is often used to prepare transparent electrodes [29]. With the continuous development of solar cells, more and more attention has been paid to the research on polymer materials and graphene (Gr) with tremendous potential in antireflection.

In the design and production process, not only are optical properties the primary factors to be considered, but mechanical properties are also key points that cannot be ignored [30]. In pursuit of better antireflection performance, the number of layers of thin film systems gradually increases and becomes more complicated. However, the available optical materials are limited, and the thermodynamic properties between them are quite different [31]. Blindly stacking the film layers would only make the whole fail faster [32]. Due to the poor antireflection effect of SLARCs and the narrower antireflection bandwidth of DLARCs, more hot spots are now focused on TLARCs and QLARCs. More layers may lead to stress mismatches. In addition, GRIN technology almost eliminates the interfaces and improves adhesion. Selecting high and low refractive index materials with opposite stress properties to form the HLHL structure could also achieve stress compensation between the layers, thereby reducing the stress of the film as a whole [13,32,33].

## 3. The Latest Applications of ARCs in the First, Second, and Third Generation of Solar Cells

### 3.1. The First Generation of Solar Cells

Crystalline silicon solar cell technology is relatively mature and has always had a high market share. In recent years, the efficiency of monocrystalline silicon and polysilicon cells has reached 26.7% and 22.3%, respectively [1]. The improvement of cell efficiency cannot be separated from the in-depth study of antireflection coatings.

Chalcogenides are abundant in nature and have good absorption and permeability. ZnS, $MoS_2$, $MoSe_2$, and other materials could be used to prepare antireflection coatings. Researchers found that colloidal metal chalcogenide quantum dots show superb quantum efficiencies in light-matter interactions and possess great device stability, and solar cells prepared with them achieved an efficiency of 11.6% under 1.5 Suns conditions [34]. The transition metal chalcogenide $MoSe_2$ has a low refractive index and a 1.57 eV band gap, which allows it to have high transparency and antireflection properties in the visible light spectrum. The $MoSe_2$ crystal structure containing niobium or other impurities has the potential for electrical and optical applications. $MoSe_2$ nanostructures were deposited on polysilicon solar cells using hydrothermal technology to study the antireflection performance by S. Santhosh et al. [35] D1 (458 nm), D2 (567 nm), D3 (761 nm), and D4 (761 nm) cell samples with different thicknesses of antireflection films were deposited on the cell surface for 30, 60, 90, and 120 min. The experimental data showed that the carrier concentration and the Hall mobility of the samples increased with a decrease in resistivity. The D3 sample had the highest external quantum efficiency (EQE), the lowest reflectivity ($\approx$7.5%), and the highest efficiency ($\approx$18.67%). The operating temperature at which the maximum efficiency was reached for each sample was also recorded, indicating that coating with ARCs would facilitate the cells to operate at high temperatures to a certain extent. Similarly, they used electric spraying technology to coat the transition metal chalcogenide ZnSe on silicon solar cells to improve cell performance. The cell sample was coated for 45 min with an ARC thickness of 1.32 μm which exhibited the maximum output efficiency and photocurrent compared with other samples [36]. Hafnium oxide ($HfO_2$) has excellent

optical properties and thermal stability, which is very suitable for preparing ARCs. Deb Kumar Shah et al. [37] used the sol-gel method to spin coat $HfO_2$ SLARCs on the surface of the textured silicon substrate at different speeds. The experimental results showed that the 70 nm layer obtained by spin coating at 2000 rpm for 30 s dropped the average reflectance from 10.51% to 6.33%. Asha Rao et al. [38] deposited 55.2 nm $TiO_2$ and 71.2 nm $Ta_2O_5$ (tantalum pentoxide) on commercial monocrystalline silicon cells. Compared with bare cells, the efficiency increased from 17.18% to 17.87% and 18.80%. Nowadays, obtaining coatings with different refractive indices by adjusting the porosity is a prominent technique. Ramakrishnan Swathi et al. [39] deposited the porous vinyltriethoxysilane/pluronic F-127 (VTES/PF-127) composite SLARC on the surface of polysilicon via spin coating at 2000 rpm for 60 s. The polymer surfactant PF-127 decomposed and left pores during the heat treatment, so the team studied the porosity, pore size, refractive index, and transmittance of the coating at annealing temperatures of 100 °C, 250 °C, and 300 °C. In the range of 400–800 nm, the maximum transmittance increased from 91.48% to 92.83%, achieving an efficiency gain of 8.75%. At the same time, the stability and hydrophobicity of the coating were further proved.

Compared with the SLARC, the DLARC could further reduce surface reflectivity and expand the bandwidth. Pariksha Malik et al. [40] grew two layers of a-$SiN_x$:H, and the stoichiometric ratio was controlled with PECVD technology, and the refractive index was adjusted accordingly. Compared with the a-$SiN_x$:H SLARC, the DLARC had lower reflectivity and higher minority carrier lifetime, $J_{SC}$, and PCE. This work shows that the multilayers of the same material could be used to reduce optical loss. Kair Kh. Nussupov et al. [41] used RF magnetron sputtering to prepare a SiC/$MgF_2$ DLARC. It was found that the magnetron power affected the atomic ratio of Si and C and the number of Si-C bonds, which in turn affected the extinction coefficient and refractive index of SiC. SiC (50 nm)/$MgF_2$ (130 nm) synthesized at a power of 100 W reduced the reflectivity to less than 3% in the range of 475–1020 nm. Rajasekar Rathanasamy et al. [42] used electrospraying technology to deposit $SiO_2$, $TiO_2$, $TiO_2$/$SiO_2$, and $SiO_2$/$TiO_2$ ARCs, and the correct deposition sequence of the two materials were experimentally proven to be $TiO_2$/$SiO_2$, which allowed the SCs to obtain a maximum efficiency gain of 30%. In the aerosol impact deposition assembly (AIDA), the synthesized $SiO_2$ nanoparticles are deposited on the glass substrate at a high impact speed, the porosity could be controlled by the time the nanoparticles remain in the reaction chamber, and the refractive index of the film could be adjusted accordingly. Kamran Alam et al. [43] used this approach to fabricate $SiO_2$ DLARCs with a porosity of 60% and 42% for the top and bottom layers, respectively, which resulted in a reduction in the reflectance from 9% to 2% over a wide range of 300–1200 nm. At the same time, the $J_{SC}$ of the polycrystalline silicon solar cells increased from 39.1 mA/cm$^2$ to 40.5 mA/cm$^2$. According to previous studies, ARCs with neutral colors have a relatively flat reflection or transmission spectra in the visible light region [44] and could create photovoltaic modules with a pure black appearance, which is of great significance in consumer electronics, architectural glass, and other fields. Yunfei Xu et al. [45] prepared a DLARC composed of top mesoporous $SiO_2$ and bottom dense $SiO_2$ using the sol-gel method. As shown in Figure 3, compared with conventional SLARCs and DLARCs, the chromaticity value is 3.35 ($C = \sqrt{a^2 + b^2}$), the data identification is closer to the center, and the color is closer to neutrality. In the visible light band of 380–780 nm, the average transmittance is 99.28%, and the PCE is increased by 2.40% when the DLARC is coated on the monocrystalline silicon micro module.

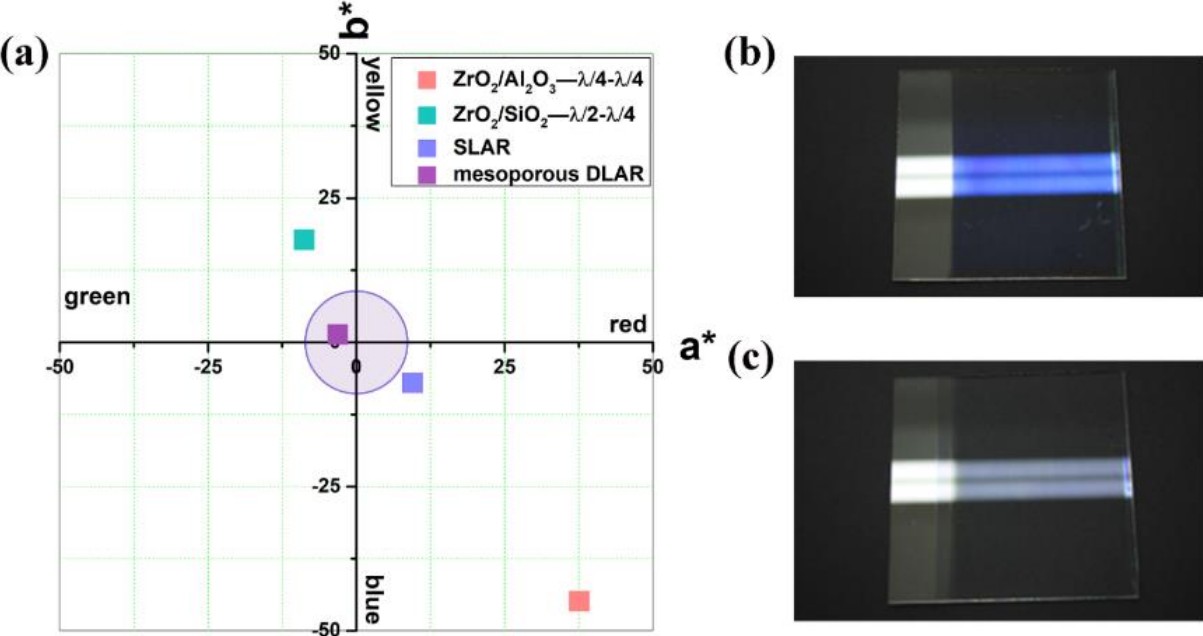

**Figure 3.** (**a**) Chromaticity points of different ARCs. a* and b* are coordinates derived from the reflectance spectrum to measure the chromaticity value. When the point falls inside the purple circle, the color is closer to neutrality. The color of the (**b**) SLARC and (**c**) mesoporous DLARC. Reprinted with permission [45]; 2020, Elsevier.

Because the DLARC could only effectively reduce the reflectance in a narrow spectral range, more and more studies have adopted three or more antireflection films to obtain lower reflectivity and wider spectral width. X.J.Xiao et al. [46] selected four materials, SiC, $HfO_2$, $SiO_2$, and $MgF_2$, and Md. Shaon Sarker et al. [47] selected three materials, ZnS, $Si_3N_4$, and $MgF_2$, who both designed and simulated three types of ARCs. The simulation results showed that the TLARC was better than the DLARC in reducing the reaction and improving cell efficiency. K. Liao et al. [48] developed and tested a novel antireflection coating ($TiO_2$-$SiO_2$/$SiO_2$/$SiN_x$) on polysilicon solar cells. The top $TiO_2$-$SiO_2$ layer, which exists in the amorphous state, was prepared with the sol-gel method, and the other two layers were deposited by PECVD. As shown in Figure 4, the top layer and $SiO_2$/$SiN_x$ DLARC are tightly bonded. Figure 4i shows a few cracks caused by the volatilization of the organic matter in the sol during annealing, which does not affect the uniformity of the ARC. Therefore, the sol-gel method is an economical method to deposit $TiO_2$-$SiO_2$ ARC effectively. The cells with the $TiO_2$-$SiO_2$/$SiO_2$/$SiN_x$ ARC have a low reflectivity of approximately 5.88% and an average PCE of 16.27% in the wavelength range of 400–1000 nm. Therefore, the antireflection stacks could efficiently decrease the light loss in a wide wavelength region, which has a good application prospect in the preparation and performance improvement. The TLARC of $SiN_x$ with different refractive indices is frequently coated on monocrystalline PERC solar cells with a PERC structure. According to the optimization scheme of D. Bouhafs [49], the first-rank refractive index of the third layer of $SiN_x$ is 1.44. It was difficult to achieve for $SiN_x$. Therefore, Shude Zhang et al. [50] replaced the third layer with $SiO_x$, with a refractive index of 1.46. When $SiO_x$ with a thickness of 10 nm replaced the $SiN_x$ with a thickness of 15 nm, the weighted average reflectivity of the two was equivalent. On the other hand, to ensure effective contact between silver and silicon, the $SiO_x$ layer should not be too thick. After comprehensive consideration, $SiO_x$ (30 nm) was selected to replace $SiN_x$ (15 nm), and the efficiency increased by 0.15%.

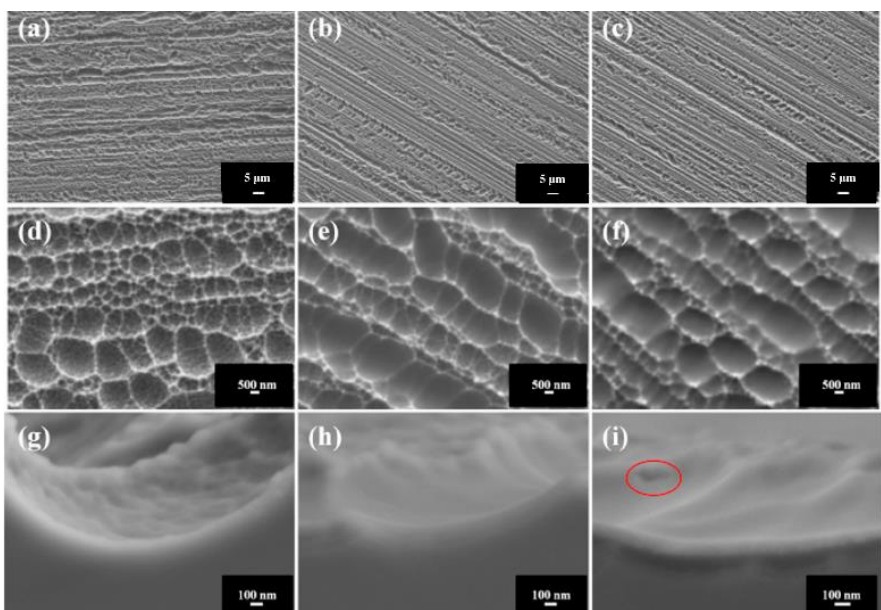

**Figure 4.** FESEM images of the polysilicon solar cells coated with: (**a**,**d**,**g**) SiN$_x$, (**b**,**e**,**h**) SiO$_2$/SiN$_x$, and (**c**,**f**,**i**) TiO$_2$–SiO$_2$/SiO$_2$/SiN$_x$ ARC. In the red circle is a crack caused by the volatilization of organic matter in the sol during annealing. The image scales of (**a**,**b**,**c**), (**d**,**e**,**f**), and (**g**,**h**,**i**) are 5 μm, 500 nm, and 100 nm, respectively. Reprinted with permission [48]; 2020, Elsevier.

For the quadruple and multiple layer ARCs, Chaoxiong et al. [51] combined a gradient index film with a multilayer ARC, and the reflectivity of the nanoporous SiO$_2$/SiO$_2$/gradient index SiO$_x$N$_y$/gradient index PS film system was simulated, which was approximately 2% in the range of 430–1200 nm. Qiang Ma et al. [52] proposed two optimization schemes where SiO$_2$ was deposited on crystalline silicon by PECVD and Al$_2$O$_3$ by DC reactive magnetron sputtering, and then SiO$_2$/SiN$_x$:H/SiN$_x$:H was added to obtain a quadruple ARC. Both the antireflection effect and PCE were improved compared to SiN$_x$:H/SiN$_x$:H DLARC. Wen-Jeng Ho et al. [53] embedded 1–3 layers of two-dimensional indium nanoparticles (In-NPs) into a SiO$_2$ antireflection film realized by electron beam evaporation and then annealed in an H$_2$ atmosphere at room temperature, which enhanced the properties of silicon solar cells by plasma scattering and the coupling effect. By measuring the reflectivity, EQE, current density, and other parameters, it was confirmed that the enhancement of J$_{SC}$ and PCE of silicon solar cells is relevant to the number of In-NPs. Compared with the bare cells, the efficiency of the cells with only the SiO$_2$ antireflection coating increased by 27.13%, while that of the cell with 3, 2, and 1-In-NPs embedded in the SiO$_2$ layer increased by 39.57%, 38.59%, and 34.27%, respectively.

Table 1 lists the experimental data of the silicon solar cells with ARCs applied in the last five years. The optical properties of the films and the efficiencies of the cells are presented in the table. ARCs mainly increase the J$_{SC}$ by reducing the loss of incident light, thereby improving cell efficiency. Regarding this point, according to the feedback of many experimental data, the V$_{OC}$ and FF of coated and uncoated cells only have a little difference which is mainly determined by the structure of the cell itself and other efficiency management methods. In three summary tables, * represents the simulated data. The improvements in optical performance and efficiency are represented by $R_{gain} = \frac{R_0 - R}{R_0}$ and $T_{gain} = \frac{T - T_0}{T_0}$, $\eta_{gain} = \frac{\eta - \eta_0}{\eta_0}$, respectively. $R_0$, $T_0$, $\eta_0$ and R, T, η denote the reflectance, transmittance, and PCE of the uncoated and coated cells, respectively. Silicon and metal based materials are still vital materials for preparing ARCs. After coating, the effect is remarkable, and some combinations could reduce the reflectivity by more than 70%. Meanwhile, compared with the SLARC, the multiple layer ARC is more efficient and stable in a wider spectrum range.

**Table 1.** The optical and electrical parameters of the first generation of solar cells with different ARCs. (* represents the simulated data.)

| Classification | Material | Optical Property (%) | Range (nm) | H (%) | $\eta_{gain}$ (%) | Reference Cell | Reference |
|---|---|---|---|---|---|---|---|
| polysilicon | MoSe$_2$ | $R_{gain}$ = 70.00 | 400–800 | 18.67 | 37.18 | bare | [35] |
| polysilicon | ZnSe | $R_{gain}$ = 70.09 | 400–800 | 19.95 | 29.80 | bare | [36] |
| crystalline silicon | HfO$_2$ | $R_{gain}$ = 39.77 | 300–1200 | 21.15 * | 72.37 * | bare | [37] |
| monocrystalline silicon | TiO$_2$ | / | / | 17.87 | 4.02 | bare | [38] |
| | Ta$_2$O$_5$ | / | / | 18.80 | 9.43 | bare | |
| polysilicon | VTES/PF-127 | $T_{max}$ = 92.83 | 400–800 | 10.56 | 8.75 | bare | [39] |
| crystalline silicon | a-SiNx:H DLARC | / | / | 17.5 | 20.69 | bare | [40] |
| crystalline silicon | SiC/MgF$_2$ | R < 3 | 475–1020 | / | / | / | [41] |
| polysilicon | TiO$_2$/SiO$_2$ | $R_{gain}$ = 70.59 | 400–800 | 18.9 | 29.99 | bare | [42] |
| polysilicon | dense SiO$_2$/porous SiO$_2$ | $R_{gain}$ = 77.78 | 300–1200 | / | / | / | [43] |
| monocrystalline silicon | dense SiO$_2$/porous SiO$_2$ | $T_{ave}$ = 99.28 | 380–780 | 20.94 | 2.40 | bare | [45] |
| polysilicon | TiO$_2$-SiO$_2$/SiO$_2$/SiN$_x$ | $R_{gain}$ = 24.23 | 300–1200 | 16.27 | 1.12 | coated with SiN$_x$ | [48] |
| monocrystalline silicon (PERC) | SiO$_x$/SiN$_x$/SiN$_x$ | $R_{gain}$ = 10.90 * | 300–1100 | 21.5 | 0.70 | coated with SiN$_x$/SiN$_x$/SiN$_x$ | [50] |
| nanostructured black silicon | nanoporous SiO$_2$/SiO$_2$/SiO$_x$N$_y$/PS | R < 2 * | 430–1200 | / | / | / | [51] |
| crystalline silicon | SiO$_2$/SiN$_x$:H/SiN$_x$:H/SiO$_2$ | $R_{gain}$ = 45.45 | 350–1100 | 14.25 | 30.38 | coated with SiN$_x$:H DLARC | [52] |
| | SiO$_2$/SiN$_x$:H/SiN$_x$:H/Al$_2$O$_3$ | $R_{gain}$ = 54.55 | 350–1100 | 14.43 | 32.02 | | |
| crystalline silicon | 1-In-NP sheet/SiO$_2$ | / | / | / | 34.27 | bare | [53] |
| | 2-In-NP sheets/SiO$_2$ | / | / | / | 38.59 | | |
| | 3-In-NP sheets/SiO$_2$ | / | / | / | 39.57 | | |
| | SiO$_2$ | / | / | / | 27.13 | | |

### 3.2. The Second Generation of Solar Cells

GaAs, the most representative III-V compound semiconductor material, has a direct band gap structure whose width is within the optimal band gap width range required by solar cell materials. It also has a large optical absorption coefficient, good radiation resistance, and a small temperature coefficient, so it is especially suitable for preparing high-efficiency space solar cells. However, GaAs is weak in mechanical strength and fragile, and it is generally fashioned into thin films to cover the substrate [15]. It was found that the following ARCs could reduce reflection and improve the efficiency of GaAs solar cells: DLC (amorphous diamondlike carbon) (50 nm) [54,55], ITO (68 nm) [56], Ta$_2$O$_5$ (67 nm) [56], ZnO (110 nm) [57] SLARC and SiO$_2$/TiO$_2$, MgF$_2$/Si$_3$N$_4$, MgF$_2$/TiO$_2$ DLARC [58] and so on. Saint-Andre et al. [59] adapted an anodic oxidation method to prepare a TiO$_2$ DLARC consisting of a dense TiO$_2$ layer at the bottom and a mixture of TiO$_2$ and voids at the top. Compared with the bare cell, the J$_{SC}$ had a brilliant change which increased by 44% in the non-packaged case and 37% in the packaged case. This shows that TiO$_2$ nanotube bilayer film has a great improvement in the performance of solar cells. Chao Ma et al. [60] adjusted the O:Al atomic ratio in Al$_2$O$_3$ by changing the working pressure in the magnetron sputtering. Figure 5 shows that the Al$_2$O$_3$ films prepared at different pressures have significantly different surface morphographies. The higher the atomic ratio, the higher the refractive index of Al$_2$O$_3$. At 2.5 Pa, the O:Al atomic ratio reached a maximum value of 2.41:1. The solar cells with the Al$_2$O$_3$/MgF$_2$ DLARC and MgF$_2$ SLARC fabricated at this pressure showed efficiency gains of 4.07% and 1.69%, respectively, compared to bare cells.

Cu (In$_{1-x}$,Ga$_x$) Se$_2$ (CIGS) material belongs to I-III-IV group of quaternary compound semiconductors, which have the advantages of a continuously adjustable band gap, a high light absorption coefficient, a powerful radiation resistance, and an excellent low light performance; thus CIGS thin film solar cells have become the research hotspot of high-efficiency photovoltaic devices [15]. A. Khadir et al. [61] found that increasing the thicknesses of ZnO (window layer) and CdS (buffer layer) would bring about a decline in PCE. It was also reported that CdS was replaced with chemical-bath-deposited ZnS(O, OH) as the buffer layer and enabled the cell to achieve an efficiency of 18.5% [62]. The researchers simulated and optimized the absorption layer thickness of the CIGS cell [61], the surface recombination rate (SRV) of CdS/CIGS, and the composition ratio of Ga/(In+Ga) [63].

Then, the effect of the MgF$_2$ SLARC with different thicknesses on the cells was studied. Dae-Hyung Cho et al. focused on the harmony between the color of photovoltaic devices and the surrounding environment. Dae-Hyung Cho et al. [64] concentrated on the harmony between the color of photovoltaic devices and the surrounding environment. They fixed the thicknesses of the Zn(O, S) buffer layer and the ITO transparent electrode, respectively, changed the thickness of the other within a certain range, and observed the color change of the cell surface under these combinations. Then, they simulated the effect of MgF$_2$ SLARC thickness and drew the conclusion that the thicker the ARC, the darker the surface color and the greater the J$_{SC}$ cell.

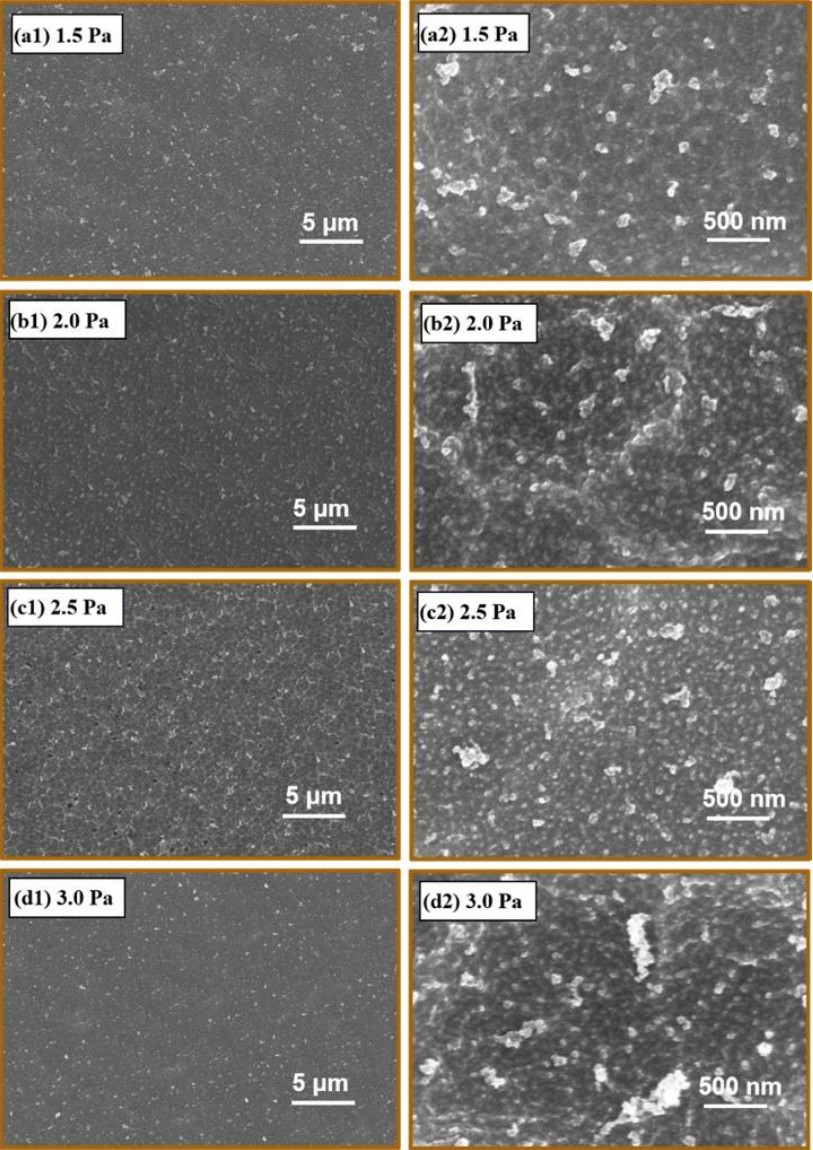

**Figure 5.** The SEM images of the surface morphology of Al$_2$O$_3$ films prepared under (**a1,a2**) 1.5 Pa, (**b1,b2**) 2.0 Pa, (**c1,c2**) 2.5 Pa, and (**d1,d2**) 3.0 Pa. (**a1**) to (**d1**) and (**a2**) to (**d2**) are the images with a 5 μm and 500 nm scale, respectively. Reprinted with permission [60]; 2021, Elsevier.

Barium titanate(BaTiO$_3$, BTO)is a perovskite material with high light transmittance in the visible light region, which was deposited on SiO$_2$ and a-Si:H/SiO$_2$ to enhance the PCE of thin film solar cells by Ľubomír Scholtz et al. [65]. The effect of transmittance enhancement is characterized by this formula. The experimental results showed that the transmittance and reflectance under the oblique incident light of 45° were significantly

better than other samples when a-Si:H/SiO$_2$ was plated with 110 nm BTO. Amorphous silicon(a-Si) solar cells are also a representative category of the second generation of solar cells. It has some outstanding elements of fewer required materials, large-area deposition, and adaptability to flexible substrates, but its conversion efficiency is relatively low at present [15]. Besides the antireflection, minimizing the spectral mismatch helps to reduce the optical loss as well. L. Meng et al. [4] used nanoporous polydimethylsiloxane (PDMS) film and perovskite quantum dot embedded composite film (PQDF) to solve these two problems. The average refractive index of the PQDF film (n$_{PQDF}$ ≈ 1.42) was between the nanoporous PDMS film (n$_{nanoporous\ PDMS}$ ≈ 1.35) and the substrate (n$_{glass}$ ≈ 1.5). Therefore, the nanoporous PDMS/PQDF DLARC not only causes the refractive index to decrease continuously from the glass substrate to the air and obtains a better antireflection effect but also effectively enhances the spectral response and realizes the downward shift of luminescence. The combination of these two films could increase the absolute PCE value of the cell from 8.95% to 10%. This photon management strategy provides a simple and effective means to solve optical problems and enhance the performance of PV devices.

The second generation of solar cells are represented by GaAs, a-Si, CIGS, and other thin film materials, which are suitable for flexible substrates. Commonly used ARCs are also applicable here. Table 2 summarizes the optical and electrical parameters of the second generation of solar cells with different ARCs. We have seen some studies on preparing double or multilayer ARCs with the same material by changing the porosity or stoichiometric ratio to adjust the refractive index. In addition to the ARCs, researchers combined the luminescence downshift film to solve the spectral mismatch and achieve better optical management. However, the overall efficiency of thin film solar cells is still relatively low. Researchers have paid more attention to the multijunction cells of these materials. Therefore, there are not many experiments and simulation studies on ARCs for thin film cells in the past three years, which need to be further deepened.

**Table 2.** The optical and electrical parameters of the second generation of solar cells with different ARCs. (* represents the simulated data.)

| Classification | Material | Optical Property (%) | Range (nm) | η (%) | η$_{gain}$ (%) | Reference Cell | Reference |
|---|---|---|---|---|---|---|---|
| GaAs | TiO$_2$ DLARC | R = 1.81 | 400–1000 | 20.6 * | 46.10 * | bare | [59] |
| GaAs | MgF$_2$ | / | / | 27.71 | 1.69 | bare | [60] |
| | Al$_2$O$_3$/MgF$_2$ | / | / | 28.36 | 4.07 | bare | |
| CIGS | MgF$_2$ | / | / | 20.52 * | 8.06 * | bare | [61] |
| CIGS | MgF$_2$ | / | / | 22.62 * | 3.29 * | bare | [63] |
| a-Si | 110 nm BTO/a-Si:H/SiO$_2$ | T$_{gain}$ = 34.4 | 400–1100 | / | / | bare | [65] |
| | 60 nm BTO/a-Si:H/SiO$_2$ | T$_{gain}$ = 18.9 | 400–1100 | / | / | bare | |
| | 112 nm BTO/SiO$_2$ | T$_{gain}$ = 17.5 | 400–1100 | / | / | bare | |
| | 61 nm BTO/SiO$_2$ | T$_{gain}$ = 15.8 | 400–1100 | / | / | bare | |
| a-Si | nanoporous PDMS/PQDF DLARC | T$_{max}$ increased by 6% (compared with nanoporous PDMS) | 380–780 | 10 | 11.73 | bare | [4] |

### 3.3. The Third Generation of Solar Cells

Organic solar cells (OSCs) belong to the third generation of solar cells with the traits of light weight, flexibility, adjustable color, and large-area printing preparation, but their efficiency is lower than others. Currently, the maximum certified efficiency of 1 cm$^2$ OSCs is 15.2% [66]. S. Kh Suleimanov et al. [67] investigated the effects of two kinds of ARCs of MgF$_2$ and CaF$_2$ mixtures with composition ratios of 95%: 5% and 55%: 45%, respectively. With the ARCs, the efficiency of the cells increased by 3.2–3.3%. Woongsik Jang et al. [68] used hollow silica nanoparticles to prepare a SLARC with a thickness of 80 nm and a refractive index of 1.10, exhibiting good broadband and omnidirectional antireflection properties. Xinjing Huang et al. [69] deposited a SiO$_2$/MgF$_2$ DLARC on a tandem OSC which used a non-fullerene acceptor with strong absorption in the near-infrared region. Compared with the uncoated cell, the cell achieved a 4.6% efficiency gain. J.-H. Kim et al. [70] spin coated a solution containing organic nanoparticles on the surface of an OSC, and when the particle size of the nanoparticles was around 86 nm, the cell

efficiency increased from 12% to 12.8%. Jingran Zhang et al. [71] synthesized and modified an epoxy resin with a high refractive index and high light transmittance. They combined the zigzag high and low refractive index resin materials through nanoimprint lithography to form a single layer composite ARC with great thermal stability (Figure 6).

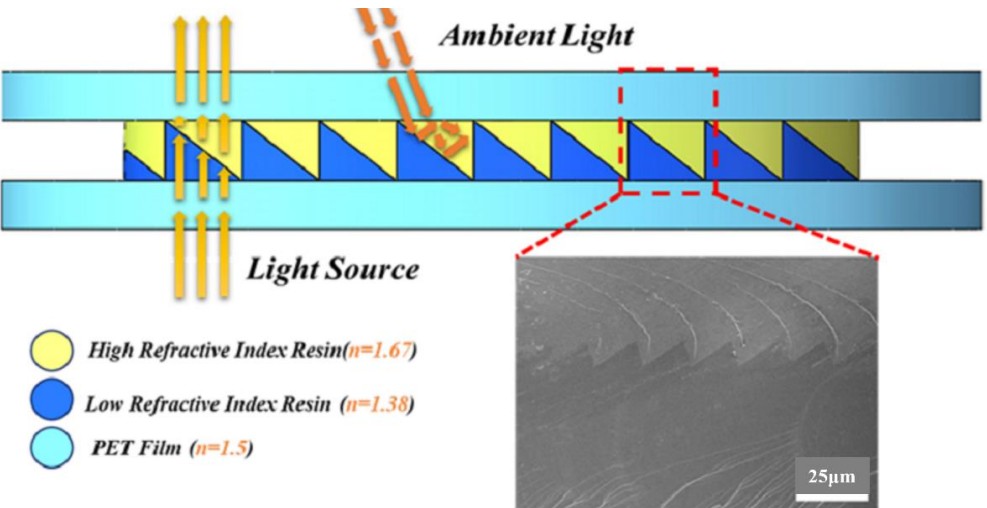

**Figure 6.** Schematic diagram of a composite SLARC consisting of high and low refractive index resins prepared by nanoimprint technology and light propagation. Reprinted with permission [71]; 2022, MDPI.

Perovskite solar cells (PSCs) have emerged in recent years, becoming a research hotspot for the advantages of high efficiency, low cost, and facile solution processability. Qi Luo et al. [72] spin coated $SiO_2$ nanospheres on cell surfaces, and the microstructure and thickness of the ARC depended on the spin coating speed. The slower the speed, the more layers of nanospheres stacked. As shown in Figure 7, when the rotational speed is 400–800 rpm, 1000 rpm, and 2000–4000 rpm, the ARCs are multilayer, sub-bilayer, and sub-monolayer, respectively. Experiments have shown that ARCs coated at 1000 rpm have the greatest gain in PCE for the nanospheres formed a pyramid-like structure, which is conducive to light trapping and reflectivity reduction. Glass texturing could effectively capture incident light and improve efficiency by light trapping structures, but it is prone to fogging and reduces transmittance. Dong In Kim et al. [73] added a mesoporous $TiO_2$/dense $TiO_2$ DLARC to it, and the PCE was improved by 24.7% relative to the bare cell and 15.3% relative to the cell with glass texturing. Flexible transparent polymer-ITO electrodes are crucial for the fabrication of flexible wearable devices. Previously, a SLARC was added to the polymer side to reduce the refractive index mismatch between the polymer and air, which was unsatisfactory. J. Zhang et al. [74] proposed a new DLARC strategy, adding a $SiO_2$ nanoparticle ARC on the polymer side and a $SnO_2$-polyethylene glycol (PEG) composite ARC on the ITO side, respectively. This method increased the average transmittance of the flexible transparent electrode from 76.8% to 89.8%. For flexible PSCs with this electrode, the efficiency was increased from 18.80% to 20.85%. Lead halide perovskite materials are easily decomposed in water, so the hydrophobic surface is pivotal to the stability and lifetime of PSCs. Mac Kim et al. [75] fabricated a PPFC (plasma-polymerized fluorocarbon)/$Nb_2O_5$/$SiO_2$/$Nb_2O_5$ (NSN) quadruple layer ARC on a polyethylene terephthalate (PET) substrate. They fixed the thickness of the NSN coating, the reflectivity and the water contact angle of the film decreased and increased gradually as the PPFC thickness increased, respectively. When the PPFC thickness is 70 nm, the PCE and water repellency are optimal.

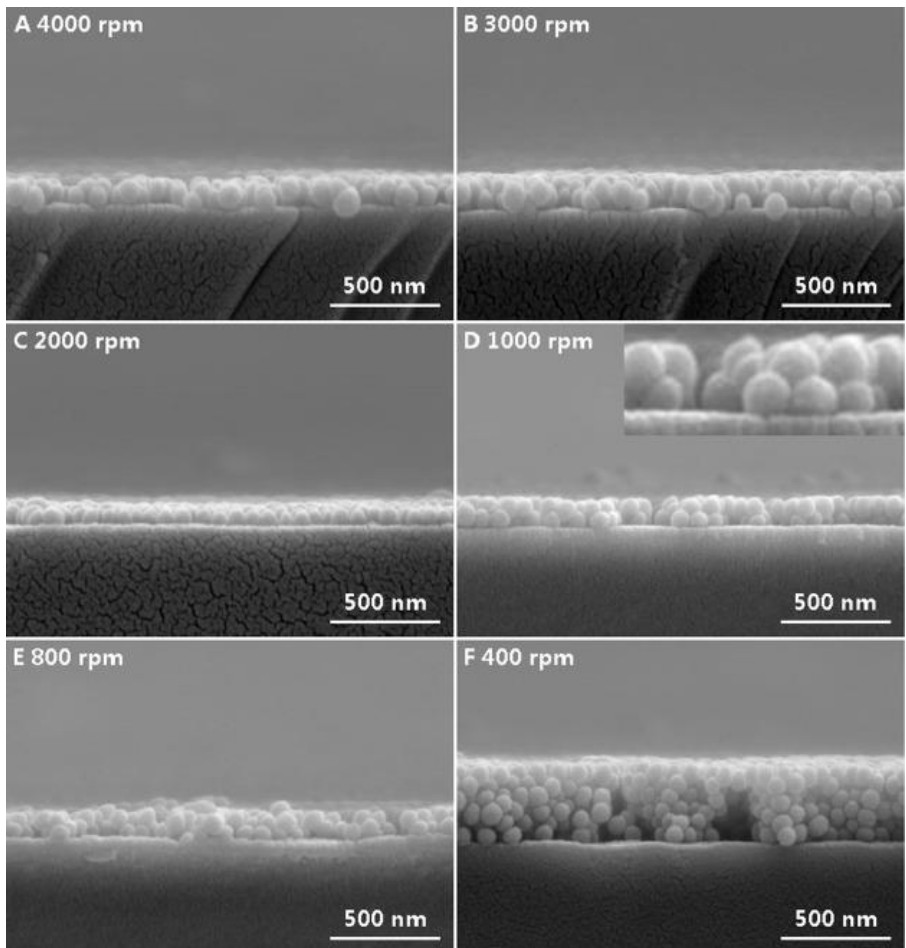

**Figure 7.** Cross sectional FESEM images of $SiO_2$ nanosphere coatings prepared at (**A**) 4000 rpm, (**B**) 3000 rpm, (**C**) 2000 rpm, (**D**) 1000 rpm, (**E**) 800 rpm, and (**F**) 400 rpm. Reprinted with permission [72]; 2018, Elsevier.

Silicon-based solar cell technology is mature, but the fabrication of the junction needs a complicated process. Graphene (Gr) has the advantages of high carrier mobility, conductivity, and optical transparency, and could be combined with a variety of semiconductors for various optoelectronic devices [76]. There is still controversy about the classification of Gr/Si cells. Based on the Schottky junction formed by Gr/Si, we classify them as heterojunction solar cells. The absorbance of single-layer Gris is low (≈2.3%), so it is difficult to obtain sufficient light. However, Gr/Si solar cells fabricated with layer transfer technology form a Schottky junction at the interface. Gr is not only a transparent top electrode but also an important part of the Schottky junction, which contributes to the separation of photogenerated carriers at the Gr/Si junction [77]. Researchers increased its efficiency from 1.65% to 15.6% in just five years since its invention [78]. $TiO_2$ [79,80], $SiN_x$ [81], and other materials are also adapted in Gr/Si solar cells. Because of the relatively high resistance of graphene, Shiqi Xiao et al. [82] prepared a coplanar nanostructure of graphene and carbon nanotubes (G-CNT) with CVD and spin coated PMMA antireflection film to increase the efficiency of the cell. Since vanadium pentoxide ($V_2O_5$) has excellent electrical and optical properties, and its refractive index is approximately 1.5–2.0, M.F. Bhopal et al. [83] used thermal evaporation technology to directly deposit $V_2O_5$ onto Gr/Si solar cells in his research. The color, current density, and electrical parameters of cells with different thicknesses of $V_2O_5$ are shown in Figure 8. From 0 to 60 nm, the color became darker and darker, which led to an increase in the current density. Since the blue color exhibited higher light absorption at a lower wavelength region of the visible spectrum, the cell coated with

a 70 nm thick film layer showed the best performance. By introducing a 70 nm $V_2O_5$ ARC, the $J_{SC}$ significantly increased by more than one time.

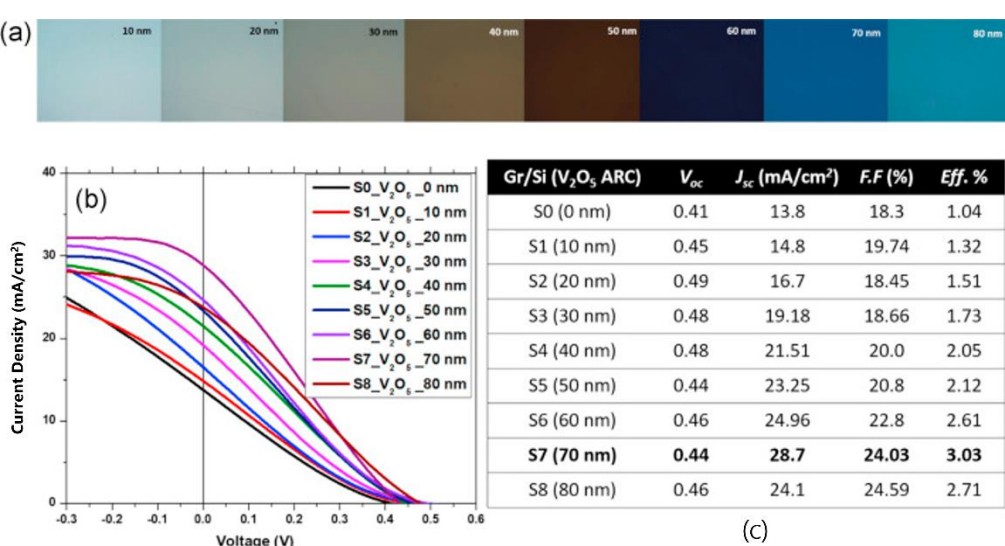

**Figure 8.** (**a**) Variation in the colors, (**b**) I-V characterization, and (**c**) electrical parameters of Gr/Si solar cells with different layer thicknesses (10–80 nm) of $V_2O_5$. Reprinted with permission [83]; 2018, Elsevier.

The structural features of silicon heterojunction solar cells (SHJs) are that a thin intrinsic amorphous silicon layer is sandwiched between c-Si and a-Si with opposite doping types [15]. SHJs have the characteristics of a simple structure, a low-temperature coefficient and process (less than 200 °C), high open-circuit voltage, and high efficiency [29]. Kunta Yoshikawa et al. [84] combined interdigitated back contact and an SHJ to achieve a PCE exceeding 26%, which is closer to the theoretical limit of silicon solar cells of 29.1%. However, the surface reflection loss would lead to a significant decrease in the photocurrent and conversion efficiency [85]. Transparent conductive oxide (TCO) thin film plays an essential role in SHJs, which is not only an ARC but also a carrier transport layer and needs to form good contact with the metal electrodes. Stanislau Y Herasimenka et al. [86] reduced the thickness of ITO by 2–3 times and deposited $SiO_x$: H on the top to form a DLARC. After annealing, due to the influence of hydrogen in $SiO_x$: H, the conductivity of ITO was improved, and the $J_{SC}$ reached 41 mA/cm$^2$. With the ITO/$SiN_x$/$SiO_x$ stacks, the $J_{SC}$ could be further increased to 42 mA/cm$^2$. Muhammad Aleem Zahid et al. [29] used a $CaF_2$/ITO DLARC to achieve a reflectivity of 6.55% in 300–1100 nm and 4.81% in 400–700 nm. Compared with traditional ITO thin films, IZO (indium zinc oxide) thin films have excellent photoelectric properties and could also be used in SHJs. Muhammad Aleem Zahid et al. [87] tried to prepare bilayer films by depositing $Al_2O_3$ on IZO with ALD, and the reflectivity in 300–1100 nm was 2.2%, showing excellent optical properties. In addition, there are many heterojunction solar cells based on other materials. Peng Xiao et al. [88] prepared an organic/silicon solar cell with a transparent graphene electrode and a DLARC, and its structure is shown in Figure 9. PMMA (polymethyl methacrylate) and PEDOT: PSS (poly (3,4-ethyleneoxythiophene): poly (styrene sulfonate) cooperate to form a DLARC, both with a thickness of 60 nm, which significantly improved the light capture ability of the silicon wafers. Under illumination, electron-hole pairs were generated at the interface of the heterojunction. Because of the built-in electric field, electrons were collected by the In-Ga back electrode and holes by the graphite electrode through the PEDOT: PSS layer. This cell was placed in the air for four months without packaging, and the efficiency of the device only attenuated by 6% compared to the original owing to the protection of the PMMA/Gr/PEDOT: PSS.

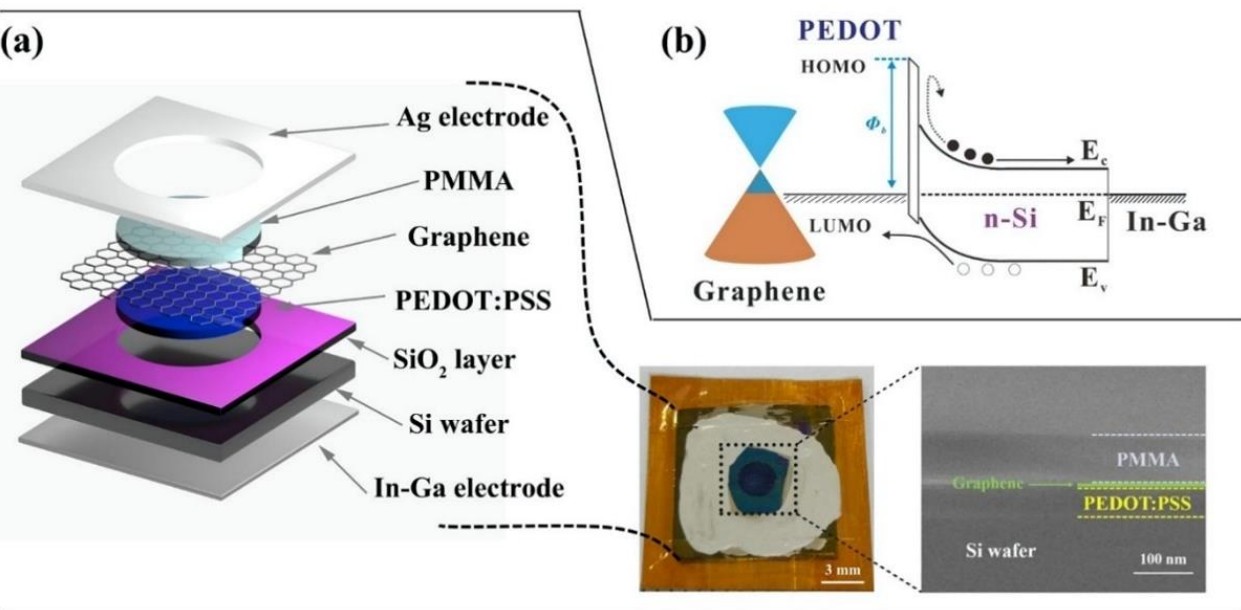

**Figure 9.** (**a**) Structure schematic and (**b**) energy band diagram of the PMMA/Gr/PEDOT:PSS/Si heterojunction solar cell. Reprinted with permission [88]; 2022, Elsevier.

Since the photoelectric response spectrum range of any single semiconductor material is narrower than the whole solar spectrum, the light energy can only be converted into electric energy within a certain range, which fundamentally restricts the improvement of efficiency. Therefore, researchers combine semiconductor materials with different band gaps to form multijunction cells. The band gap of the top material is the largest, which decreases in turn, and absorbs light in different wavelength ranges, which could greatly improve the utilization rate of sunlight. Among the third generation of solar cells, multijunction solar cells tend to achieve higher efficiency than the Shockley-Queisser limit of single cells because they absorb a wide solar spectral range and minimize heat loss [89]. Furthermore, the advantages of high reliability, high power-to-mass ratio, and excellent radiation intensity render them more suitable for space applications. A. A. El Amin et al. [90] studied a $TiO_2/SiO_2$ DLARC for $CdTe_{0.65}P_{0.35}$/Si dual-junction solar cells, which could achieve a reflectivity below 5% in the wavelength range of 300–1100 nm. S. B. Musalinov et al. [91] designed and optimized a $TiO_2/SiO_2$ DLARC and $TiO_2/Si_3N_4/SiO_2$ TLARC for InGaP/GaAs/Ge triple junction cells. The results showed that the TLARC significantly increased the photocurrent density of the Ge subcell, thereby contributing to the overall performance of the whole cell. Tae Soo Kim et al. used an ultrathin $SiO_2$ layer [92] and $Al_2O_3/SiO_2$ [93] as hydrophobic and antireflection layers on the surface of dual-junction InGaP/GaAs cells, and the accelerated life test results showed that the life of encapsulated cells in water at room temperature was greatly extended. R. Campesato et al. [94] designed a TLARC ($Nb_2O_5/Ta_2O_5/SiO_2$) and GRIN films based on $Nb_2O_5$ for triple junction solar cells, which both benefited the antireflection. Although the antireflection performance of the GRIN film at the wavelength of 300–1800 nm was outstanding, it mainly played a role in the infrared region of the bottom junction, which could not significantly contribute to the overall cell efficiency. Weinan Zhang et al. [95] prepared a $TiO_2/Al_2O_3/MgF_2$ TLARC on GaInP/InGaAs/Ge triple junction solar cells by electron beam evaporation, which decreased the average reflectivity from 25.57% to 3.37% in the wavelength range of 300–1800 nm. Thus, the cell absorbed more photons and produced more photocurrent. However, the ARC could not affect the internal resistance; the fill factor, mainly determined by the parallel resistance and series resistance inside the cell, was almost unchanged [96]. As shown in Figure 10, according to the concept of the equivalent layer, replacing the intermediate refractive index layer in the middle of the TLARC with the equivalent HLH layer, an HLHL quadruple layer

ARC could be formed [13]. Similarly, G. Hou et al. [33] designed high and low refractive index stacks (HLIS) to realize the antireflection. Two groups of materials, $MgF_2/ZnS$ and $Al_2O_3/TiO_2$, were selected in the experiment. Firstly, the optimum thicknesses of the two groups as the DLARC were discovered. The resulting DLARC was cut into a stack of equal thickness (2.5 nm or 5 nm) thin layers, and the researchers swapped one target layer at a time, and the change was retained if the performance was better after flipping. Through continuous iteration, the optimal result was found (the process is depicted in Figure 11). The effects of the DLAR and HLIS on the $J_{SC}$ of GaInP/Ga(In)As/Ge cells and retrograde GaInP/GaAs/GaInAs/GaInAs cells were simulated, which show that HLIS has better omnidirectional properties, and a possible method to prepare the HLIS was proposed.

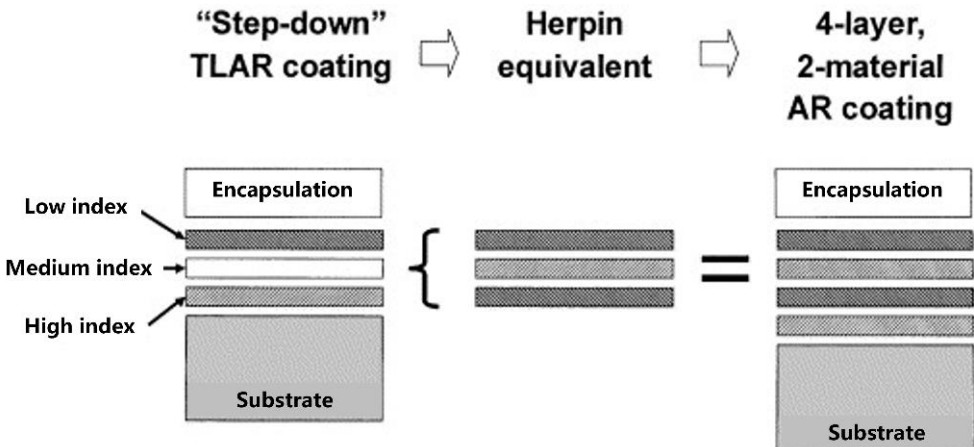

**Figure 10.** Scheme of the HLHL ARC structure derived from a TLARC structure using the concept of the equivalent layer. Reprinted with permission [13]; 2000, Elsevier.

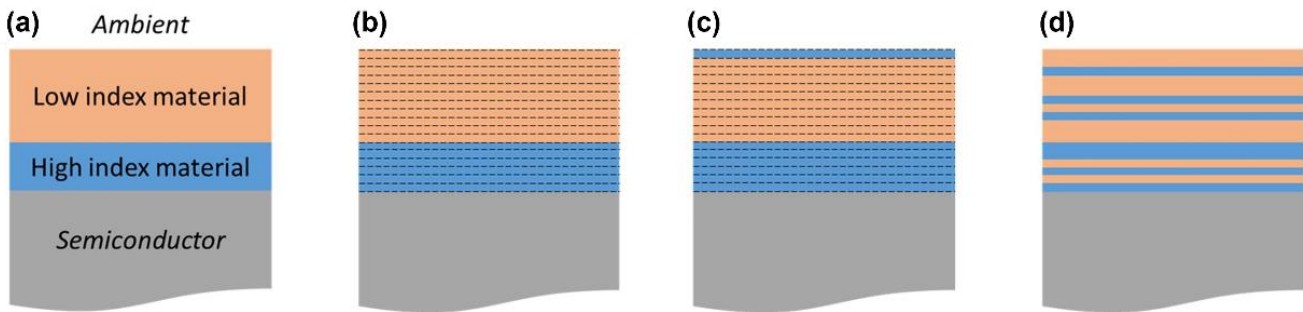

**Figure 11.** The design process of the HLIS film: (**a**) selecting the optimum thickness of the high and low refractive index materials, (**b**) cutting the DLARC into layers of equal thickness, (**c**) swapping one target layer at a time, and (**d**) forming the HLIS after several iterations. Reprinted with permission [33]; 2021, Elsevier.

Owing to these advantages, researchers have conducted much research on this type of cell. Table 3 shows that more composite and polymer coatings appear and possess great antireflection and durabilities. Apart from the DLARC with the different porosity, we also noticed the research of combining and assembling high and low refractive index materials to form a flat ARC via nanoimprint technology, which enhanced the omnidirectional antireflection property. The HLHL structure could reduce the kinds of materials while achieving better antireflection effects, which means that researchers do not need to explore more combinations of antireflection materials. However, a better way is needed to stack multiple layers by turns. On the other hand, the GRIN structure could greatly reduce the refractive index mismatch between the incident medium and the substrate, but is limited by the current preparation technology; there are few studies in this area. Of

course, there are many ways to achieve antireflection performance besides optical films. Gurjit Singh et al. [97] embedded Al nanoparticles with a radius of 80nm under a $Ta_2O_5$ ARC, using plasma resonance and antireflection to boost the light absorption rate and photocurrent generation of the cell. F. Chaudhry et al. [98] coated nanoparticles with dielectric properties on the top of GaAs solar cells to alleviate the reflectivity and guide the incident light into the active layer more effectively. On the other hand, it avoided the geothermal effect of plasma nanoparticles and reduced the harm to the cell life. In addition, textured surfaces, such as moth-eye antireflection nanostructures [99,100], could be designed with efficient photothermal conversion and spectral absorption coatings for photovoltaic applications by selecting appropriate materials and appropriate deformation depths. This is also a current research hotspot.

**Table 3.** The optical and electrical parameters of the third generation of solar cells with different ARCs. (* represents the simulated data.)

| Classification | Material | Optical Property (%) | Range (nm) | $\eta$ (%) | $\eta_{gain}$ (%) | Reference Cell | Reference |
|---|---|---|---|---|---|---|---|
| OSC | $MgF_2$-$CaF_2$ mixture (55:45) | | | | $\Delta\eta = 3.2$ | bare | [67] |
| | $MgF_2$-$CaF_2$ mixture (95:5) | | | | $\Delta\eta = 3.3$ | | |
| OSC | Hollow $SiO_2$ NPs | R > 90 | 420–1000 | 6.53 | 8.83 | bare | [68] |
| OSC | $SiO_2$/$MgF_2$ | R decreased by 4% | 400–1000 | 15.9 | 4.61 | bare | [69] |
| OSC | Poly(styrene-co-acrylic acid) nanoparticle | $T_{max} = 98.9$ | 400–800 | 12.8 | 6.67 | bare | [70] |
| OSC | high-low RI resin | $T_{max} = 99.86$ | 300–800 | / | / | / | [71] |
| PSC | $SiO_2$ NPs(1000 rpm) | $T_{max} = 96.1$ | 400–800 | 15.82 | 6.82 | bare | [72] |
| PSC | dense $TiO_2$/mesoporous $TiO_2$ | $T_{max} \approx 80$ | 300–1200 | 13.95 | 24.78 | bare | [73] |
| PSC | $SiO_2$ NPs +$SnO_2$-PEG | $T_{gain} = 16.93$ | 400–800 | 20.85 | 10.90 | bare | [74] |
| PSC | PPFC(70nm)/NSN/HC-PET | $R_{min} = 1.71$ | 400–1200 | 17 | 5.59 | coated with HC-PET | [75] |
| G-CNT/Si | PMMA | / | / | 9.1 | 19.74 | bare | [82] |
| Gr/Si | $V_2O_5$ | $R_{min} = 2$–3 | 300–1000 | 3.03 | 191.35 | bare | [83] |
| SHJ | $CaF_2$/ITO | $R_{gain} = 31.70$ | 300–1100 | 21.05 | 2.93 | coated with ITO | [29] |
| SHJ | $Al_2O_3$/IZO | 4.9 | 300–1100 | 21.57 | 3.85 | coated with IZO | [87] |
| Organic/Si | 60 nm PMMA/1-Gr/60 nm PEDOT:PSS | / | / | 13.01 | 25.46 | coated with 60 nm PEDOT:PSS | [88] |
| Dual-junction InGaP/GaAs | $SiO_2$ | / | / | 28.77 | 2.24 | bare | [92] |
| dual-junction InGaP/GaAs | $Al_2O_3$/$SiO_2$ | / | / | 27.4 | 6.30 | bare | [93] |
| GaInP/InGaAs/Ge TJ | $TiO_2$/$Al_2O_3$/$MgF_2$ | $R_{gain} = 86.82$ | 380–1800 | 32.71 * | 29.44 * | bare | [95] |

## 4. Conclusions

Optical reflection loss is a crucial factor restricting the efficiency improvement of solar cells. This paper briefly introduces the transfer matrix method in optical thin films, which is the basic method and principle of designing single, double, and multiple layer ARCs. Then, starting with the first, second, and third generation of solar cells, the latest research on ARCs in the past three years from the experimental aspect is introduced. More novel antireflection materials have emerged in recent years, while typical materials and combinations still have a wide range of applications. Meanwhile, more researchers pay attention to other measures, such as increasing the porosity, adjusting the stoichiometric ratio, combining the quantum dots, and adding nanoparticles or nanosheets. In addition to reducing optical loss, ARCs are also endowed with more expectations, such as hydrophobicity, self-cleaning, and neutral colors.

It is worth noting that refractive index matching is a key point, which determines the optical performance of the film system. However, the consideration of stress matching between the materials is rarely reflected in the design process, which determines the mechanical properties of the film system. The internal stress derives from the growth mode of the film and the microstructure, and the thermal stress is the force generated during the temperature change due to the difference in the thermal expansion coefficient. They act together on the film-substrate system. Excessive stress would cause problems, such as fracture, warping, and delamination of the film, resulting in failure. Although heat treatment techniques, such as thermal annealing, could greatly eliminate or change

film stress, related issues should be considered at the beginning of the design. The classic coatings that repeatedly appear in the research of ARCs must have considered the stress matching problem at the beginning and accepted many tests.

**Funding:** This research was supported by the National Key R&D Program of China (No. 2019YFB1503602), the Strategic Priority Research Program of the Chinese Academy of Sciences (No. XDB43020500), and the Scientific Instrument Developing Project of the Chinese Academy of Sciences (No. YJKYYQ20200002).

**Institutional Review Board Statement:** Not applicable.

**Informed Consent Statement:** Not applicable.

**Data Availability Statement:** Not applicable.

**Conflicts of Interest:** The authors declare no conflict of interest.

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
