# Peer review of "Recent Applications of Antireflection Coatings in Solar Cells"

_photonics, doi:10.3390/photonics9120906_

Round 1
Reviewer 1 Report
The review article by Ji et al. summarizes some of the recent research on anti-reflective coatings (ARCs) for solar cells. It starts off by going through the different design principles for an ARC (single, double, triple layer, gradient reflective index coatings, and equivalent layers). In the second part, it gives examples on the use of ARCs in 1st, 2nd, and 3rd generation solar cells.
The utilization of ARCs in solar cells is important in order to achieve high efficiencies, and there is a lot of research going on at the moment. However, two recent reviews articles from 2020 summarize in more detail the latest developments in the field (https://doi.org/10.3390/en13102631 and https://doi.org/10.1016/j.solener.2020.01.084, both of which are cited in the current manuscript). Compared to these reviews, I do not get much added information from the current paper. The first part discussing the design principles, is well-known to the community, while the second part mainly highlights some examples from the field (not only recent discoveries). Thus, I have difficulties to understand why this review is needed. At least this should be clearly stated in the introduction.
In my opinion, I think the focus of the manuscript should be improved and my recommendation is that it needs major revision before it can be considered to be published in Photonics. Detailed comments below.
1. Most of the time, the language is good. However, some of the words used by the authors are clearly wrong. For instance, the authors use the word “batteries” repeatedly for something which I believe should be “devices”. Another example is “plasma scattering” (e.g. on line 346). This should most likely be “plasmon” or “plasmonic” scattering. There are also other small mistakes throughout the manuscript, which should be corrected.
2. Concerning the design principles of ARCs, Chapter 2 does not contain many references, although this theoretical part should be well-described in the literature. As mentioned earlier, this should be well-known to the community. On the other hand, this part cannot be excluded either.
3. HLHL is mentioned in the abstract. However, the abbreviation is not explained (neither in the abstract nor in the manuscript).
4. Figure 3 shows the structure of the devices studied in ref [17]. However, I think the description given by the authors on lines 212-219 does not really explain the concept of these devices. This description should be improved.
5. Table 2 shows the device performances of a range of 1st generation solar cells with different ARC technologies. In my opinion, this does not give much useful information as the performance of the devices without the ARC is not given (the efficiencies of these are also varying and a comparison cannot be made). As the focus of this review is on the ARC and not the solar cell itself, it would be better to compare the improvement in device performance the various ARC technologies can provide. Furthermore, in Table 2, very diverse Voc, Jsc and FF values can be seen. However, the authors do not reflect on these at all (which I think is good, because it is more related to the solar cell below the ARC than on the ARC itself).
6. According to Table 3, the solar cell in ref [4] has a Voc of 2.54 V. I think for an a-Si solar cell this is not possible.
7. There is a bit of debate on which solar cells belong to which generation of solar cells. In my opinion, the Gr/Si and CNT/Si solar cells mentioned in Chapter 3.3 would still be classified as first generation, because of their similarities to c-Si solar cells. However, this is of course debatable. When it comes to tandem and multijunction solar cells, it is more accepted that these belong to the third generation. It should still be pointed out that, although very efficient, the production costs of multijunction cells are still very high.
8. The organic solar cell part is very short (only 6 lines) and could be easily expanded. Another very promising third generation solar cell technology not mentioned at all in the paper is perovskite solar cells. I think it would good to also include the latest work on ARCs for perovskite solar cells.
9. As mentioned above, I think the second part of the review just shallowly gives examples on various ARCs for different solar cells. However, it would good if the authors also could give some more insights on why different ARCs should be used for various types of solar cells. Or should they? What are the design criteria to be considered?
Reviewer 2 Report
The manuscript can be published after a minor revision
In this paper, authors review the latest applications of antireflection optical thin films in different types of solar cells, basic optical theories of designing antireflection coatings, commonly used antireflection materials, and their classical combinations are introduced.
I have some comments about the manuscript:
-Some keywords should be removed
- Line 32, it is not necessary to explain what is a solar cell
-There are many errors in the references. For example, in the citation
19, the pages are missing (38….). The authors should check all references.
-Line 246, figure 4 is too small.
-It is necessary to incorporate the name of the figures 4a/5a and 4b/5b in the text.
- The authors should include in the captions of the figures the corresponding permission from each editorial. If they do not have the permission, they can't use them.
-Line 322, figure 6 is too small, it is impossible to read the scale
- In general, the authors should not remove information from the original captions of some figure.
- The number of the chemical formula should be in subscripts. Several of these errors have been found: line 324 (SiO2 instead of SiO2) , 325, 35, 473…..
- In captions of the figure 8, the authors should include the description of a and b, separately
-Line 472, figure 12 a is too small and the letter c is missing in the figure
-Line 561, in the figure 16, the letters a, b, c, d are missing in the captions.
- In the Figure 16, the authors changed the figure and removed the word “ambient”, they should include it.
Round 2
Reviewer 1 Report
Based on the cover letter, I think the authors have addressed the concerns I had in the first version of the manuscript. However, it was difficult to read the final version of the manuscript due to the "track changes" function. However, I trust that the changes are OK.
Thus, in my opinion, the manuscript can be accepted for publication. However, I recommend that a thorough proof reading should be carried out prior to publication.